# Lessons Learned from the Lessons Learned in Public Health during the First Years of COVID-19 Pandemic

**DOI:** 10.3390/ijerph20031785

**Published:** 2023-01-18

**Authors:** Alessia Marcassoli, Matilde Leonardi, Marco Passavanti, Valerio De Angelis, Enrico Bentivegna, Paolo Martelletti, Alberto Raggi

**Affiliations:** 1Neurology, Public Health, Disability Unit and Coma Research Center, Fondazione IRCCS Istituto Neurologico Carlo Besta, 20133 Milan, Italy; 2Department of Clinical and Molecular Medicine, Sapienza University, 00189 Rome, Italy

**Keywords:** COVID-19, public health, lessons learned, World Health Organization, risk factors, digital technologies, coordination

## Abstract

(1) Objectives: to investigate the main lessons learned from the public health (PH) response to COVID-19, using the global perspective endorsed by the WHO pillars, and understand what countries have learned from their practical actions. (2) Methods: we searched for articles in PubMed and CINAHL from 1 January 2020 to 31 January 2022. 455 articles were included. Inclusion criteria were PH themes and lessons learned from the COVID-19 pandemic. One hundred and forty-four articles were finally included in a detailed scoping review. (3) Findings: 78 lessons learned were available, cited 928 times in the 144 articles. Our review highlighted 5 main lessons learned among the WHO regions: need for continuous coordination between PH institutions and organisations (1); importance of assessment and evaluation of risk factors for the diffusion of COVID-19, identifying vulnerable populations (2); establishment of evaluation systems to assess the impact of planned PH measures (3); extensive application of digital technologies, telecommunications and electronic health records (4); need for periodic scientific reviews to provide regular updates on the most effective PH management strategies (5). (4) Conclusion: lessons found in this review could be essential for the future, providing recommendations for an increasingly flexible, fast and efficient PH response to a healthcare emergency such as the COVID-19 pandemic.

## 1. Introduction

On 30 January 2020, the SARS-CoV-2 outbreak and its associated disease (COVID-19) were declared a “Public Health emergency” by the WHO. On 6 February 2020, the WHO and the UN Development Coordination Office published the COVID-19 Strategic Preparedness and Response Plan (SPRP), with the aim of responding to the emergency as “One United Nation” [1]. The SPRP outlined public health (PH) measures as “action plans” to immediately support countries in being prepared to respond to the SARS-CoV-2 2019 pandemic; these action plans provided practical guidelines with priority steps for countries’ PH management during the COVID-19 emergency.

COVID-19 SPRP was developed across the major areas of PH response, including services and activities that could be undertaken to manage health at the population level.

To accompany the COVID-19 SPRP, the WHO published Operational Planning Guidelines [1,2,3], with the aim of responding to countries’ PH organisational needs. Since then, the WHO has regularly updated these guidelines to follow up on the evolution of the virus and the scientific knowledge about it.

Three global guidelines were published until 31 January 2022 [1,2,3], with updates on the main “action plans” defined across different PH areas or “pillars”: for example, the vaccination pillar was added during the course of the guideline update. The third revision (effective from 1 February 2021 to 31 January 2022) includes 10 pillars [3].

In addition to global guidelines, some WHO regions developed their own operational planning guidelines to respond to region-specific situations, e.g., Strategic Preparedness and Response Plan for the WHO African region [4] or the COVID-19 Strategic Preparedness and Response Plan in the WHO’s Eastern Mediterranean region [5]. Since 11th March 2020, the global COVID-19 pandemic was officially declared by the WHO, and almost all countries have had to face its consequences. In May 2021, the WHO reported five main lessons learned through its Monitoring and Evaluation (M&E) Framework [6], which sets out the approach and methods for tracking and reporting global progress against the COVID-19 pandemic. Such lessons are however organised as generic broad principles rather than targeted concrete actions and therefore do not provide a vision of what has been carried out in single contexts or countries, but are shaped as general recommendations.

To understand which lessons were learned by healthcare workers, politicians and communities about the PH management of COVID-19 from countries around the world, there was a need to capture the real-world perspective.

The objective of this scoping review is to investigate the main and most relevant lessons learned by all countries in the following WHO regions: African region (AFRO), Eastern Mediterranean region (EMRO), European region (EURO), region of the Americas (AMRO), South-East Asia region (SEARO), and Western Pacific region (WPRO). We evaluated the world’s public health responses to COVID-19 from around the world, using the WHO pillars as a reference framework. This review is also aimed at understanding whether there is consistency between what the WHO planned for countries’ PH management and what countries have learned from the practical actions undertaken during the first 2 years of the COVID-19 pandemic.

## 2. Materials and Methods

### 2.1. Search Strategy

Our work has been performed following PRISMA (Preferred Reporting Items for Systematic Reviews and Meta-Analyses) guidelines.

PubMed and CINAHL electronic databases were searched to identify studies published in English between 1 January 2020 and 31 January 2022 that examined the lessons learned in PH during the first two years since the beginning of the SARS-CoV-2 pandemic. In each database, the search terms for studies’ titles/abstracts combined the term “Lesson * Learned” with the following keywords: COVID */, Coronavirus/, SARS-CoV-2, Public Health. See Box 1 for the full search strings.

Box 1Key terms used to identify studies on lessons learned in public health management during COVID-19 pandemic(COVID *[Title/Abstract] OR Coronavirus[Title/Abstract] OR SARS-CoV-2[Title/Abstract])AND(Lesson * learned[Title/Abstract])AND(“public health”[Title/Abstract])*: This symbol indicates the truncation, which includes all the possible variables connected to that name e.g. “Lesson/Lessons” or “COVID/COVID-19”.

### 2.2. Inclusion and Exclusion Criteria

Studies were included if they: (1) were original reports; (2) addressed lessons learned in management of the COVID-19 pandemic, dealing with one of the 10 pillars reported by the WHO in its most recent Operational Planning Guidelines (WHO, 2021), namely (i) coordination, planning, financing, and monitoring; (ii) risk communication, community engagement, and infodemic management; (iii) surveillance, epidemiological investigation, contact tracing, and adjustment of public health and social measures; (iv) points of entry, international travel and transport, mass gatherings and population movement; (v) laboratories and diagnostics; (vi) infection prevention and control, and protection of the health workforce; (vii) case management, clinical operations, and therapeutics; (viii) operational support and logistics, and supply chains; (xi) strengthening essential health services and systems; (x) vaccination.

Studies were instead excluded if: (1) were not in English; (2) did not involve human beings (e.g., animal models); (3) relied on lessons learned from other pandemics or diseases to inform COVID-19 management; (4) the information could not be referred to one or more of the WHO pillars; (5) article type was a review, conference proceeding or editorial; and (6) the full text was not available (for full-text analysis only).

### 2.3. Manuscripts Selection

Once duplicates were removed, retained records were screened based on title/abstract by four researchers independently (MP, AM, VDA, EB). To increase the quality and consistency of data extraction, 20% of the abstracts assigned to each reviewer were selected for a second check by another one of the four researchers, who was blinded to the first decision. In case of disagreement, the study was retained for full-text reading. If the agreement rate was below 70%, another 20% set of abstracts was submitted to double-check.

Full texts of retained records were checked for selection criteria. Reviewers were supervised by three experienced scientists (ML, PM, AR) during the whole selection phase: in case of difficulty with the designation of the final outcome on the paper, i.e., whether it had to be included or not, reviewers referred to one of them for a second opinion.

### 2.4. Data Extraction and Synthesis

We designed a data extraction form to record data from the studies, including, which journal, year, authors, title, geographic area of the study (WHO regions), and participants’ information (number and mean age) were reported. In this form, the data have been divided into 10 main sections, one for each WHO pillar [4]. In each section, a definition of the public health area covered by the pillar, the total number of articles linked to the pillar, the total number of available lessons connected to each pillar, as well as the total number of references to each lesson learned within each pillar (which constitutes the main result) were included. Some studies might contain lessons referring to multiple WHO regions; thus, they were double-counted for each region mentioned.

A qualitative data synthesis approach was followed to describe the main lessons. For each pillar, we presented the most frequently reported lessons learned across all WHO regions: for operational purposes, we set a 30% threshold referring to the number of occurrences in which a specific lesson learned was assigned within each pillar. For each of these most frequent lessons, we reported an example taken from one paper.

Finally, as we aimed to address the most relevant lessons, we decided to identify key public health lessons learned by relying on a practical approach, i.e., that they were crosswise present within more than half of the 10 pillars.

## 3. Results

From the initial 455 studies, 144 were finally included for the qualitative synthesis [7,8,9,10,11,12,13,14,15,16,17,18,19,20,21,22,23,24,25,26,27,28,29,30,31,32,33,34,35,36,37,38,39,40,41,42,43,44,45,46,47,48,49,50,51,52,53,54,55,56,57,58,59,60,61,62,63,64,65,66,67,68,69,70,71,72,73,74,75,76,77,78,79,80,81,82,83,84,85,86,87,88,89,90,91,92,93,94,95,96,97,98,99,100,101,102,103,104,105,106,107,108,109,110,111,112,113,114,115,116,117,118,119,120,121,122,123,124,125,126,127,128,129,130,131,132,133,134,135,136,137,138,139,140,141,142,143,144,145,146,147,148,149,150], conforming to PRISMA guidelines (Figure 1). The agreement rate at abstract check was 92%, whereas for 16 studies (i.e., 8.5% of retained full texts), a second opinion was sought.

A total of 78 lessons learned have been found, which were cited 928 times in the 144 papers. The majority of papers and lessons were from AMRO (75 papers and 360 single lessons). Figure 2 shows the 928 citations of the 78 lessons learned across the six WHO regions.

### 3.1. Pillar 1: Coordination, Planning, Financing, and Monitoring

The first WHO pillar is about PH coordination and multi-sectoral/multi-partner management of countries’ responses to COVID-19, including the engagement of different and relevant national ministries [3].

A total of 11 lessons, derived from 77 studies [7,8,9,10,11,12,13,14,15,16,17,18,19,20,21,22,23,24,25,26,27,28,29,30,31,32,33,34,35,36,37,38,39,40,41,42,43,44,45,46,47,48,49,50,51,52,53,54,55,56,57,58,59,60,61,62,63,64,65,66,67,68,69,70,71,72,73,74,75,76,77,78,79,80,81,82,83] and cited 190 times, were referred to this pillar; among them, 25 citations referred to AFRO region, 20 to EMRO region, 20 to EURO region, 79 to AMRO region, 29 to SEARO, and 17 to WPRO region (see Appendix A).

Three lessons covered 30% or more of the total amount of lessons learned: (1) coordination between healthcare and other organisations (28 citations); (2) coordination between local and regional public health institutions (24 citations); (3) conduct initial capacity assessment and risk analysis, including mapping of vulnerable populations (21 citations). One example for each of them is reported in Table 1.

### 3.2. Pillar 2: Risk Communication, Community Engagement, and Infodemic Management

Pillar 2 is about community risk communication and engagement, regarding what is known about COVID-19, what is still unknown, what is being done and the PH actions to be taken [3].

A total of 16 lessons, derived from 43 studies [10,12,14,19,22,27,28,29,30,31,34,36,41,45,48,50,52,54,58,62,63,65,67,72,76,80,84,85,86,87,88,89,90,91,92,93,94,95,96,97,98,99,100] and quoted on 178 single citations, were referred to this pillar: among them, 12 citations were referred to AFRO region, 28 to EMRO region, 22 to EURO region, 67 to AMRO region, 28 to SEARO, and 21 to WPRO region (see Appendix A).

Four lessons covered at least 30% of the total amount of lessons learned: (1) Communication effectiveness and social needs evaluation (20 citations); (2) which use of information and communication technologies (17 citations); (3) availability of rapid response services (16 citations); (4) development of knowledge translation platforms (16 citations). One example for each of them is reported in Table 1.

### 3.3. Pillar 3: Surveillance, Epidemiological Investigation, Contact Tracing, and Adjustment of Public Health and Social Measures

Pillar 3 is about surveillance objectives during the COVID-19 pandemic: promoting rapid detection of imported cases, comprehensive and rapid contact tracing, and case identification [3].

A total of 17 lessons, derived from 47 studies [8,9,10,12,15,16,20,21,22,25,26,27,29,30,31,36,37,43,44,47,48,49,55,57,58,60,70,72,73,74,75,77,80,81,87,101,102,103,104,105,106,107,108,109,110,111,112] and quoted on 167 single occasions, were referred to this pillar: among them, 30 citations referred to AFRO region, 16 to EMRO region, 25 to EURO region, 52 to AMRO region, 29 to SEARO, and 15 to WPRO region (see Appendix A).

Two lessons covered at least 30% of the total amount of lessons learned: (1) digital technologies and electronic health records (31 citations); (2) healthcare coalitions and lines of communication (31 citations).

One example for each of them is reported in Table 1.

### 3.4. Pillar 4: Points of Entry, International Travel and Transport, Mass Gatherings and Population Movement

Pillar 4 is about surveillance and risk communication activities at countries’ entry points/borders during the COVID-19 PH emergency [3].

A total of 5 lessons, derived from 13 studies [10,12,20,31,43,47,57,65,70,72,87,99,113] and quoted on 19 single occasions, were referred to this pillar: among them, 2 citations referred to AFRO region, 4 to EMRO region, 3 to EURO region, 1 to AMRO region, 6 to SEARO, and 3 to WPRO region (see Appendix A).

A single lesson covered at least 30% of the total amount of lessons learned: restrictive travel rules allow better control of the pandemic (6 citations). An example is reported in Table 1.

### 3.5. Pillar 5: Laboratories and Diagnostics

Pillar 5 is about countries’ laboratory capacity to manage large-scale testing for COVID-19, including the adoption and dissemination of standard operating procedures for collection, management, and transportation of COVID-19 diagnostic testing [3].

A total of 6 lessons, derived from 27 studies [7,10,12,16,22,34,39,43,44,49,59,60,68,70,73,75,77,81,85,107,114,115,116,117,118,119,120] and quoted on 53 single occasions, were referred to this pillar: among them, 10 citations were referred to AFRO region, 5 to EMRO region, 3 to EURO region, 24 to AMRO region, 7 to SEARO, and 4 to WPRO region (see Appendix A).

A single lesson covered at least 30% of the total lessons learned: national level coordination on data search and sharing (23 citations). An example is reported in Table 1.

### 3.6. Pillar 6: Infection Prevention and Control, and Protection of the Health Workforce

Pillar 6 is about infection prevention and control practices in communities and health facilities to prevent transmission and prepare for COVID-19 patients’ treatment, and to prevent transmissions to surge staff and to all patients/visitors [3].

A total of 7 lessons, derived from 59 studies [10,12,14,15,16,17,20,22,25,30,31,36,37,45,47,48,51,55,62,64,65,69,70,72,73,74,76,79,80,81,82,83,84,86,88,90,102,106,107,108,111,112,113,116,121,122,123,124,125,126,127,128,129,130,131,132,133,134,135] and quoted on 107 single occasions, were referred to this pillar: among them, 17 citations were referred to AFRO region, 8 to EMRO region, 16 to EURO region, 40 to AMRO region, 14 to SEARO, and 12 to WPRO region (see Appendix A).

Two lessons covered at least 30% of the total amount of lessons learned: (1) attention to the poorest countries, low-income people and vulnerable patients in terms of aid and support (23 citations); (2) how IT and telemedicine help provide PH support (21 citations). One example for each of them is reported in Table 1.

### 3.7. Pillar 7: Case Management, Clinical Operations, and Therapeutics

Pillar 7 is about suspected COVID-19 case definition and case management. The WHO underlines that guidance should be made available on how to manage different patients with, or at risk of, severe illness, and mild severe cases, having special considerations for vulnerable populations too (such as the elderly, people with chronic diseases, pregnant or lactating women, and children) [3].

A total of 7 lessons, derived from 29 studies [9,10,11,12,13,20,27,28,31,45,55,64,67,70,72,73,75,78,84,85,87,111,112,122,134,136,137,138,139] and quoted on 50 single occasions, were referred to this pillar: among them, 8 citations were referred to AFRO region, 4 to EMRO region, 10 to EURO region, 15 to AMRO region, 11 to SEARO, and 2 to WPRO region (see Appendix A).

A single lesson covered at least 30% of the total amount of lessons learned: Build protocols in case-management to cover all areas (15 citations). An example is reported in Table 1.

### 3.8. Pillar 8: Operational Support and Logistics, and Supply Chains

Pillar 8 is about operational support and logistical arrangements to support incident management and operations, e.g., surge staff deployments, procurement of essential supplies, staff payments [3].

A total of 10 lessons, derived from 61 studies [7,8,10,11,12,14,15,16,17,19,20,23,25,28,30,33,39,43,45,48,49,50,51,54,55,57,58,60,63,65,67,71,72,73,78,82,83,85,91,96,101,116,118,121,122,123,129,131,136,137,138,140,141,142,143,144,145,146,147,148,149] and quoted on 121 single occasions, were referred to this pillar: among them, 19 citations were referred to AFRO region, 2 to EMRO region, 15 to EURO region, 64 to AMRO region, 9 to SEARO, and 12 to WPRO region (see Appendix A). Two lessons covered at least 30% of the total amount of lessons learned: (1) increase and repurpose existing equipment, laboratories and physical space (29 citations) and (2) prepare staff surge capacity and deployment mechanisms (23 citations). One example for each of them is reported in Table 1.

### 3.9. Pillar 9: Strengthening Essential Health Services and Systems

Pillar 9, added in the WHO Operational Planning Guidelines of January 2021 [3], is about the need to strengthen essential health services and systems during a global public health crisis. In the middle of the COVID-19 pandemic, countries all around the world experienced difficulties in the balance of their healthcare response to coronavirus disease, still maintaining safe delivery of other essential healthcare services. The WHO underlined that the prioritisation and selection of essential health services depend on the health system’s baseline capacity, the socio-economic conditions of the population, the COVID-19 transmission context, and the burden of disease [3].

A total of 8 lessons, derived from 16 studies [10,11,27,28,55,67,75,78,85,87,112,122,134,136,137,150] and quoted on 23 single occasions, were referred to this pillar: among them, 3 citations were referred to AFRO region, 2 to EMRO region, 3 to EURO region, 10 to AMRO region, 4 to SEARO, and 1 to WPRO region (see Appendix A). The lesson “look at healthcare management through a global perspective” emerged from one paper and was intended as a global-level recommendation. Two lessons covered at least 30% of the total amount of lessons learned: (1) use of technologies and telemedicine to deliver healthcare (6 citations); (2) multi-professional and collaboration teamwork (5 citations). One example for each of them is reported in Table 1.

### 3.10. Pillar 10: Vaccination

Pillar 10, added in the WHO Operational Planning Guidelines of January 2021 [3], is about COVID-19 vaccines and WHO recommendations about early planning, regulation, policy, communications, logistics and distribution of vaccines, including attention to national deployment and vaccination plans [3].

A total of 10 lessons, derived from 10 studies [17,28,76,80,81,92,109,115,116,124] and quoted on 18 single occasions, were referred to this pillar: among them, 3 citations were referred to AFRO region, 2 to EMRO region, 5 to EURO region, 6 to AMRO region, none to SEARO, and 2 to WPRO region (see Appendix A).

Two lessons covered at least 30% of the total amount of lessons learned: (1) prioritise available vaccine stocks appropriately based on risks and vulnerabilities (3 citations); (2) an electronic health record (EHR) platform can support structured data to maximise discrete data entry and vaccination tracking (3 citations).

One example for each of them is reported in Table 1.

### 3.11. Key Public Health Lessons Learned in the WHO Regions

A total of five lessons were reported within six or more of the pillars and were therefore addressed as key lessons (see Box 2). The first was “Need for continuous coordination” which was cited in 8 pillars (pillars 1, 2, 3, 5, 6, 7, 9, 10): this focuses on supporting PH and healthcare responses, ensuring risk analysis, mapping vulnerable populations and early identifying outbreaks. The second was “Importance of assessment end evaluation” which was cited in 7 pillars (pillars 1, 2, 3, 4, 6, 8, 9): This is aimed at assessing the effectiveness of planned measures, mapping knowledge gaps in the population and aligning research and evidence synthesis topics with policy needs. The third was “Establishing evaluation systems” which was cited in 6 pillars (pillars 1, 3, 7, 8, 9, 10), and which should help facilitate rapid and evidence-informed responses during crises, thus prioritising relevant issues. The fourth was “Extensive application of digital technologies” which was cited in 9 pillars (pillars 1, 2, 3, 4, 6, 7, 8, 9, 10): this focuses on increasing efficiency and security of investigation and contact tracing, making it easier to produce reliable data, improving communication between stakeholders and clinicians, and, finally, regularly informing the population on the ongoing pandemic. The fifth was “Need for periodic scientific reviews” which was cited in 6 pillars (pillars 1, 3, 5, 8, 10), and which is aimed to produce summary information about different aspects of COVID-19 (or other) pandemics, e.g., the need for testing, incidence, outcomes—including, for example, hospitalisation rates, or frequency of post-infection sequelae—and finally forecast the overall population health once the waves are over.

Box 2Key common public health lessons learned in the WHO regions
Need for continuous coordination, updating and communication between public health institutions and organisations at the local, national and global level.Importance of assessment and evaluation of risk factors for the diffusion of COVID-19, by developing risk analysis to identify vulnerable populations.Establishing evaluation systems to assess the impact of planned public health measures and evaluate which are functional and which have not led to improvements in the epidemiological and/or healthcare situation.Extensive application of digital technologies, telecommunications and electronic health records to update and communicate with the population, improve case surveillance and management, enable coordination between politics and healthcare institutions, and provide online medical services.Need for periodic scientific reviews to maintain regular updates on the most effective and functional public health management strategies adopted by different countries all over the world.


## 4. Discussion

In this study, we investigated the main lessons learned from public health management of the COVID-19 pandemic, covering the period from 1 January 2020 to 31 January 2022, and retrieved from 144 papers a total of 78 lessons learned, cited 928 times in all the papers, and five key lessons. Four are the main results of our review.

First, there were considerable differences in the distribution of lessons learned by countries among the WHO pillars [3], i.e., across different public health areas, with most of the lessons coming from the areas of coordination of activities, management of the information to the general population, and surveillance and contact tracing.

Second, lessons were unevenly distributed among the different WHO regions: we observed clear trends in their geographical distribution, with the highest number of publications on lessons learned from the AMRO region; meanwhile, AFRO published several lessons learned on the exploitation of available resources to develop new public health strategies to guarantee essential services during the pandemic. The region with the lowest number of published lessons learned was WPRO, which is one of the most densely populated and the one from which SARS-CoV-2 was spread around the world.

Third, we found a close association between some of the lessons learned from different countries and the WHO pillars guidelines [3], although other relevant public health guidelines were not found in this review.

Finally, we found that there were some lessons that addressed, in a transversal way, different themes of public health and different geographical areas, while other lessons were specific, for specific countries and public health systems and services.

The distribution of the lessons differs significantly between pillars [3]: the frequency varies from a minimum of five lessons learned in pillar 4 to a maximum of 17 lessons learned in pillar 3. Similarly, the number of citations to the lessons was quite variable, with the first three pillars showing the largest number of citations. It is interesting to note that the first three pillars—country-level coordination, planning, monitoring; risk communication and community engagement; surveillance, rapid response teams, and case investigation—were referred to in the initial phases of the pandemic, when countries needed strong guidance from the WHO. Conversely, the vaccination pillar (no. 10) was the one with the lowest number of citations; in our opinion, this is due to the fact that, given the time frame of our review, too little time has passed from the organisation and implementation of vaccination plans to the identification of lessons learned and finally to their publication.

The 10 pillars of the WHO Operational Planning Guidelines [3] have been a roadmap to classify lessons learned into different public health areas. With such an approach, we can also compare the action plan outlined by the WHO in its guidelines at the beginning of the pandemic with what has been actually implemented in practice by countries. Our results show a noteworthy proximity between the actions defined in each of the 10 pillars and the public health lessons learned by countries until January 2022. For example, it is interesting to notice that the WHO action plan “Integrate and continue to promote a ‘whole-of-society’ approach to coordination, specifically to position the health sector response within the broader socioeconomic response and recovery” in pillar 1 matches the lesson learned by countries: “Coordination between healthcare and other organisations” [3]. Another example is pillar 2, where the action “Monitor the communication and community engagement actions that aim to facilitate trust and population adherence to public health measures” [3] matches with the lesson learned: “Communication effectiveness and social needs evaluation”.

The WHO reported more actions that were not found in this review, although they could represent additional strategies to be implemented in PH action plans of importance in times of emergencies. For example, the WHO recommended in pillar 1 to “Develop initiatives to reduce out-of-pocket payments and other financial barriers to access health care and essential health services, and mitigate the effect of movement restriction (e.g., free or subsidised access to telecommunications, food supplies, social and income protection)” and in pillar 7 to “Ensure comprehensive medical, nutritional, psycho-social, and palliative care for those with COVID-19” [3]. Although we could not find any lesson related to these actions in our review, that does not mean that they are not relevant: reporting of lessons learned from the COVID-19 pandemic represents a very minor part of the COVID-19 literature, and therefore the possibility that lessons are still to be clearly identified as connected to specific actions is a possibility that should not be excluded.

We understand that it is difficult to obtain generalizable lessons across different public health settings and contexts, ignoring national diversities; however, as already stated, some of the lessons learned identified in our work clearly address different themes of global public health importance and cover different geographical areas, thus showing their global relevance. We identified five key lessons that were crosswise referred to by the majority of pillars, and that thus represent key cornerstones for any public health strategy in emergencies.

The key lessons provide important advice in order to manage the current course of the pandemic and the future implications of COVID-19 or other diseases. These lessons teach health policies, systems and services to constantly pay attention and update on health emergencies all over the world, in order to rapidly develop efficient and pragmatic responses to help other countries and to prevent the possibility of disease’s spread. During a pandemic, one of the most important actions to be taken is the identification of the most vulnerable populations; rapid preventive measures for these populations will help to decrease the spread and mortality of the disease itself. If a public health decision worsens the progress of the pandemic or is ineffective, the development of systems able to quickly assess the impact of planned public health measures could provide a rapid and efficient evaluation and correction of the actions taken. Our work highlights that the scientific community has to develop periodic scientific reviews on the most effective and functional public health management strategies implemented at national and international levels, so as to learn from each lesson and improve future planning in emergencies. New technology-based hardware and software are fundamental tools in many areas of public health management during a global crisis: electronic platforms for communication improved coordination among stakeholders and early communication to support the PH and healthcare response to COVID-19 diffusion. Technology tools, such as the application of mobile phones, can automate or accelerate aspects of case investigation and contact tracing, but also could be used to check COVID-19 symptoms and train first-contacts on the measures needed to avoid infections [106], thus reducing the required workforce. Shared online and interactive databases, frequently updated with new data, allow users to interact with healthcare and public health data and create analyses and visualisations on the ongoing pandemic. Technology also helped increase the use of certified electronic health records around the world and supervise COVID-19 vaccination through the so-called “green pass” policy [80].

We have to acknowledge some limitations. The first is related to the search string itself, which mostly focused on the two main terms “Lessons Learned” and “Public Health”. Although these terms allowed the authors to focus their work on a more specific number of articles, they may have caused the exclusion of papers containing lessons learned in public health but which did not explicitly declare them as above. Second, a literature search was performed excluding all languages but English and grey literature, thus missing all national or unpublished guidelines and lessons. Another issue, which is not properly a limitation, but a note or attention to the readers, is referred to as “pillar 10”, “Vaccination”: Although this pillar represents a fundamental public health issue in managing the pandemic, in our review it included a limited number of articles and lessons learned. Probably, the time needed to generate one or more lessons about COVID-19 vaccination was likely not enough to adequately represent vaccination-related lessons learned in the first 24 months.

## 5. Conclusions

In conclusion, after analysing the public health responses to the COVID-19 pandemic from countries around the world, 78 lessons were learned by countries 2 years after the beginning of the pandemic. The WHO regularly provides support to countries by updating and developing strategic operational public health guidelines [1,2,3]. In our review, out of these 78, we highlighted five key overarching public health lessons, not all of which are described in the WHO recommendations for the public health management of COVID-19. We believe that these five main global lessons cover fundamental public health areas that should be considered for present and future emergencies by countries as well as by international bodies working in emergencies and which constitute key public health strategies in times of healthcare crisis.

The five lessons could be complemented at international and particularly at national levels also by guidelines more in line with principles of equity of access to care as well as of the Universal Health Coverage approach, as suggested in the WHO COVID-19 Operational Planning Guidelines but not found in the lessons published so far [3].

Although it is impossible to precisely predict how the COVID-19 pandemic will evolve, it is necessary to continue to work, maintaining and implementing what has been previously learned about COVID-19 surveillance and management, in order to become increasingly vigilant, flexible, fast and efficient at local, national and global levels. For these reasons, all the lessons found in the present review should be known and the most relevant ones should be shared and implemented so as to provide a more solid and structured global public health response.

Coordination, communication, surveillance, transport, diagnostics, prevention, and logistics are the essential pillars should an emergency occur. We learned and are learning in this COVID-19 era that the main lesson is that all these pillars should be considered when planning to forecast, prevent and face global emergencies.

## Figures and Tables

**Figure 1 ijerph-20-01785-f001:**
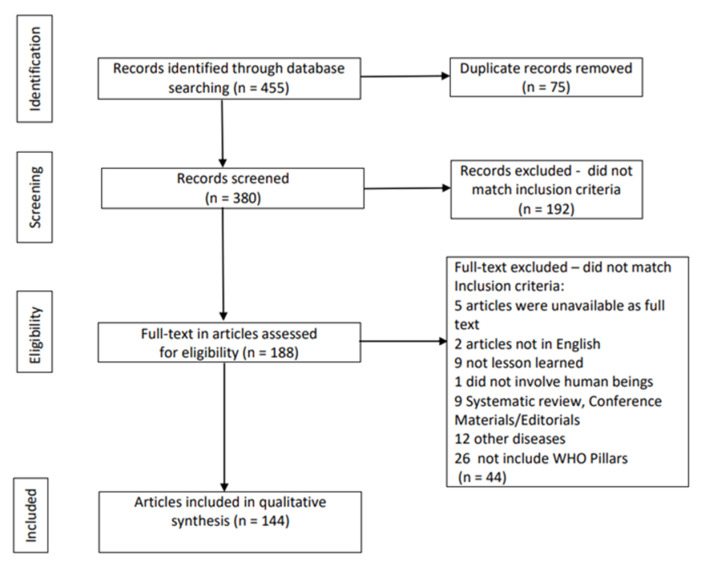
Flow diagram of studies’ selection, conforming to PRISMA guidelines.

**Figure 2 ijerph-20-01785-f002:**
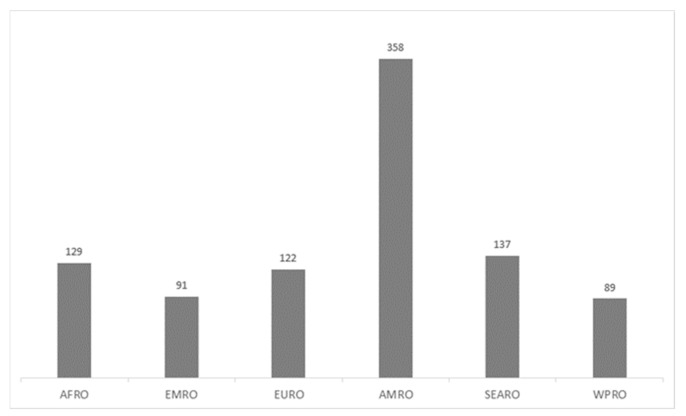
The 928 citations of the 78 lessons learned across WHO regions.

**Table 1 ijerph-20-01785-t001:** Practical examples from most reported lessons learned in the 10 pillars (from WHO COVID-19 Operational Planning Guidelines).

Pillar	Main Lessons Learned	Example	Country (Region)	Reference
Pillar 1: Coordination, planning, financing, and monitoring	1.1. Coordination between healthcare and other organizations	Local governments in Sri Lanka, under the supervision of the provincial director of health services, developed an awareness programme from their collaboration with the local medical officer of health and public health inspectors to increase knowledge at a community level and to promote local autonomy and preparedness. The authors learned that a strong and free healthcare, public health and community care collaborative system is necessary to combat a health crisis.	Sri Lanka (SEARO)	Hettiarachchi et al., 2021 [41]
	1.2. Coordination between local and regional public health institutions	The healthcare coalition of Western Washington created a Western Washington Regional COVID Coordinating Centre, functioning as a coordinating cell, established a network across local healthcare systems (which usually worked in a competitive landscape). The authors learned that such a strategy ensured that none of the hospitals was overburdened, thus maintaining effective care and capacity for all patients.	Washington state (US) (AMRO)	Mitchell et al., 2020 [55]
	1.3. Conduct initial capacity assessment and risk analysis, including mapping of vulnerable populations	Development of an estimation method for the state-level case fatality rate in India, also investigating its associated factors to develop a map of vulnerable populations and targeted PH interventions. The authors learned that a state-level case fatality rate enabled them to address the main risk factors for the transmission of COVID-19, which specifically for this context included poverty, health inequalities and poor socio-economic status.	India (SEARO)	Asirvatham et al., 2020 [15]
Pillar 2: Risk communication, community engagement, and infodemic management	2.1. Communication effectiveness and social needs evaluation	Assessment of information understanding among travellers arriving at the major UK ports in the early stages of the pandemic. Questionnaire survey and follow-up interviews involved 121 passengers, assessing their knowledge of symptoms, actions and attitudes towards PH information. The authors learned that questionnaires and interviews can be useful tools to evaluate public understanding and feedback about COVID-19 public communications.	United Kingdom (EURO)	Zhang et al., 2021 [99]
	2.2. Which use of Information and Communication Technologies	Development of “COVID-19 Conversations” programme by Northwell health, New York’s largest healthcare provider. The programme delivered interactive Zoom/Facebook Live discussions between experts and population on COVID-19 topics, which also addressed mental, physical and psychological needs of the community. The authors learned that easy online access and culturally relevant community education and outreach represent efficient interventions to precisely inform the public in a timely, accurate, and comprehensive way about COVID-19	State of New York (AMRO)	Williams et al., 2022 [97]
	2.3. Availability of rapid response services	The authors illustrated the rapid development of a knowledge translation platform in Lebanon, which had the capability and flexibility to rapidly respond to different stakeholders’ demands during a public health crisis, by tailoring its services to address various kinds of requests. The authors learned that the development of knowledge translation platforms can help to reduce the time limitations during a PH crisis, providing timely evidence to decision-makers, which have to urgently meet PH needs.	Lebanon (EMRO)	El-Jardali et al., 2020 [34]
	2.4. Development of knowledge translation platforms	Knowledge translation platforms in Lebanon facilitated and coordinated the communication between policy and practice, bringing together stakeholders from different sectors, e.g., policy-makers, scientists, healthcare workers, or civil society organizations to share and increase mutual understanding and challenges, also providing a neutral real-time communication channel for population.	Lebanon (EMRO)	El-Jardali et al., 2020 [34]
Pillar 3: Surveillance, epidemiological investigation, contact tracing, and adjustment of public health and social measures	3.1. Healthcare coalitions and lines of communication	The authors described the Monitoring and Evaluation plan adopted by 35 counties in the WHO AFRO region; these countries activated coordinated data management and progress reports, which were compiled and analysed by WHO AFRO to oversee the national pandemic situation and response. Thirty-five countries submitted daily or weekly line lists with detailed epidemiological reports, expressed in Key Performance Indicators. The main lesson that the authors learned was that a supranational Monitoring and Evaluation plan allowed WHO regions to identify and supervise priority counties in need of support.	35 Countries of Africa (AFRO)	Impouma et al., 2021 [43]
	3.2. Digital technologies and Electronic Health Records	Anne Arundel County Department of Health (Maryland) adopted in its contact tracing activities a National Electronic Disease Surveillance System: positive COVID-19 test results were reported electronically and a nurse staff created case files, assigning each one to a team member for daily monitoring and follow-up. The main lesson that the authors learned was that an online contact-tracing platform allows greater standardization of data collection and ease in monitoring of case patients.	Maryland (US) (AMRO)	Kalyanaraman and Fraser, 2020 [104]
Pillar 4: Points of entry, international travel and transport, mass gatherings and population movement	4.1. Restrictive travel rules allow better control of the pandemic	The Vietnamese government applied a series of flight suspensions right after the first COVID-19 confirmed case: from March 5, 2020, the Country suspended and denied any foreign flights and Vietnamese people around the world were also suggested not to return to Vietnam if not necessary. The main lesson that the authors learned was that prompt action, since the very early stages of the pandemic, allowed to contain the infections from the points of entry of a limited-resource country.	Vietnam (WPRO)	Duong et al., 2020 [31]
Pillar 5: Laboratories and Diagnostic	5.1. National level coordination on data search and sharing	Description of the creation and evolution of a rapid review service in Canada, built by the National Collaborating Centre for Methods and Tools on internationally accepted rapid review methodologies, with the objective to address primary COVID-19 PH questions. The authors learned that rapid review methods can ensure coordination, knowledge and implementation of the best available evidence and practice among laboratories all over the country during a PH crisis.	Canada (AMRO)	Neil-Sztramko et al., 2021 [59]
Pillar 6: Infection prevention and control, and protection of the health workforce	6.1. Attention to the poorest countries, low-income people and vulnerable patients in terms of aid and support	Singapore adopted a PH approach that involved the whole society, with a specific focus on the support of foreign workers, a marginalised and economically vulnerable population forming most of the infected and hospitalized people, also due to their overcrowded accommodations and limited healthcare access. The authors learned that multiple measures must be undertaken to prevent and control the diffusion of COVID-19 infections, such as promoting suitable housing environments, raising healthcare literacy or providing economic benefits to encourage social distancing and lockdown compliance among the population.	Singapore (WPRO)	Wang and Teo, 2021 [81]
	6.2. The IT and telemedicine help to provide PH support	Hospital Pharmacy Service in Spain adopted telepharmacy services to promote remote outpatient pharmacy care, teleconsultation with drug dispensing and home drug delivery. The authors underlined the usefulness of telepharmacy services for clinical follow-up, healthcare coalitions and coordination, outpatient counselling, and dispensing and delivery of medication during the COVID-19 pandemic, also as a new and complementary tool for ordinary healthcare consultation.	Spain (EURO)	Margusino-Framiñán et al., 2020 [127]
Pillar 7: Case management, clinical operations, and therapeutics	7.1. Build protocols in case management to cover all areas	The American College of Academic International Medicine and the World Academic Council of Emergency Medicine reported on the importance of building case management protocols that involved daily communication of established best practices for emergency and critical care, ensuring clinical strategies and resources and instructing personnel. The authors learned that data-driven and applied protocols built on best practices, established early on during the pandemic, can improve therapeutic outcomes and reduce the need for endotracheal intubations and intensive care unit utilisation.	United States of America (AMRO)	Stawicki et al., 2020 [73]
Pillar 8: Operational support and logistics, and supply chains	8.1. Increase and repurpose existing equipment, laboratories and physical space	During the first phases of the COVID-19 pandemic the Ethiopian Public Health Institute rapidly built a new COVID-19 testing laboratory from its available resources. Ethiopian Public Health Institute identified appropriate spaces, renovated them using national WHO guidelines, and mobilised underutilised resources (equipment, materials and consumables), mainly from the Malaria and Neglected Tropical Diseases research team. The authors learned that with a strong commitment, teamwork, leadership and strong management support, new COVID-19 testing laboratories can be established, also in resource-limited settings.	Ethiopia (AFRO)	Abera et al., 2020 [7]
	8.2. Prepare staff surge capacity and deployment mechanisms	In addition to taking measures to protect the members, Canadian Armed Forces have contributed to Canada’s response to COVID-19 by developing training for allied health professionals and ensuring adequate resources to manage staff surge capacity. The authors learned that political and workforce engagement and provisional relocation mechanisms (which involved, in this case, the national army) were necessary to prepare rapid and strong PH response to COVID-19.	Canada (AMRO)	Edge et al., 2020 [33]
Pillar 9: Strengthening essential health services and systems	9.1. Use of technologies and telemedicine to deliver healthcare	Six healthcare workers’ focus groups were created by the researchers to discuss across various working frontline in PH management of COVID-19; integration of information technology support to speed up and maintain essential healthcare services delivery was reported by all the stakeholders as a key lesson learned from the healthcare management of COVID-19 pandemic.	Oman (EMRO)	Al Ghafri et al., 2020 [11]
	9.2. Multi-professional and collaboration teamwork	The authors reported that a key lesson learned during the COVID-19 pandemic was about the importance of close cooperation between hospital Emergency Departments (ED) and the department of infectious diseases. The split of ED into multiple teams was considered essential. The Department Chair, as the ED Commander, gave operational command and control over five clinical operational teams and forward screening area (FSA) teams.	China (WPRO)	Quah et al., 2020 [138]
Pillar 10: Vaccination	10.1. Prioritize available vaccine stocks appropriately based on risks and vulnerabilities	In Saudi Arabia, a distribution plan for vaccines was developed based on a phased approach that identified priority populations by estimating, for example, the size of targeted populations, the assurance of essential healthcare services continuity, or the population greatest at risk in the community. The authors learned that, when the vaccine supply was not enough to provide coverage for all the population, a phased distribution plan helped to overcome the challenges of vaccine supply.	Saudi Arabia (EMRO)	Assiri et al., 2021 [17]
	10.2. Electronic health record (EHR) platform can support structured data to maximise discrete data entry and vaccination tracking	The authors described an example of EHR platform use to maximise vaccination tracking, especially for USA patients with spina bifida: the development of a standardised health registry enabled easier reporting and collection of individual health records, cross-care coordination, and vaccinations updating during the COVID-19 pandemic.	United States of America (AMRO)	Castillo et al., 2021 [124]

## Data Availability

Not applicable.

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
