# Peer review of "Lessons Learned from the Lessons Learned in Public Health during the First Years of COVID-19 Pandemic"

_ijerph, 2023, doi:10.3390/ijerph20031785_

Round 1

Reviewer 1 Report

This manuscript provides a nice review for the public health response to COVID-19 pandemic from countries around the world, 78 were the lessons learned by countries after two years from the beginning of the pandemics. Five key overarching public health lessons are highlighted, not all described in the WHO recommendations for the public health management of COVID-19. I agree with the authors that these five main global lessons cover fundamental public health areas that should be considered for present and future emergencies by countries as well as by international bodies working in emergencies and which constitute key public health strategies in times of healthcare crisis.  

Overall, this review paper is well written. I suggest minor revision with the following suggestions.

Suggestion 1: The authors are suggested to discuss the benefits of using new technology-based hardware and software like mobile apps in healthcare during Covid-19. 

Suggestion 2: The authors are suggested to give a summary of the data sets used in the reviewed papers. If some are publicly available, it is also suggested to give URLs of them.

Reviewer 2 Report

Interesting paper. The approach typical for systematic literature reviews conforms to acceptable standards regarding search strategy and inclusion/exclusion criteria. It is an important topic that connects WHO pillars for fighting pandemics with the classification of papers containing “lessons learned” from practice. Good classification of papers to mentioned pillars, although in the Results section, the description should be more detailed – including some findings from analysed papers beyond simple classification, or at least better reference supplementary material. The paper is well referenced as it is required for SLR-type papers.
